# Test Anxiety and Related Factors among Health Professions Students: A Saudi Arabian Perspective

**DOI:** 10.3390/bs12040098

**Published:** 2022-04-08

**Authors:** Abdulaziz Alamri, Muhammad Ashraf Nazir

**Affiliations:** Department of Preventive Dental Sciences, College of Dentistry, Imam Abdulrahman Bin Faisal University, Dammam 31441, Saudi Arabia; absalamri@iau.edu.sa

**Keywords:** health sciences students, exam anxiety, stress, academic performance

## Abstract

The study aimed to evaluate test anxiety and its relationship with demographic factors among undergraduate medical, dental, and pharmacy students in Dammam, Saudi Arabia. The health professions students from Imam Abdulrahman Bin Faisal University, Dammam, Saudi Arabia participated in this cross-sectional study. Test Anxiety Inventory (TAI) by Spielberger was used to evaluate test anxiety and the score on the scale ranges from 20 to 80. Of the total 1098 participants, 878 returned completed questionnaires and the response rate of the study was 79.9%. In response to the items of TAI, 33% of participants reported that they almost always “wished examinations did not bother so much”. The mean TAI score of the sample was 43.17 (SD = 10.58). There were statistically significant differences in the mean scores of TAI among dental (44.15 ± 0.48), medical (41.64 ± 1.31), and pharmacy (43.44 ± 9.29) students (*p* = 0.003). The students with high grade point average (GPA) (mean TAI = 41.91 ± 10.43) demonstrated significantly lower test anxiety than those with low GPA (mean TAI = 44.05 ± 10.67) (*p* = 0.006). Academic grade in the previous year (GPA) remained a statistically significant factor associated with test anxiety (B = −2.83, *p* = 0.003) in multivariate analysis. This study showed that test anxiety was common among health professions students, and dental students and students with low GPA demonstrated high test anxiety. Students with high test anxiety should be the target of preventive strategies.

## 1. Introduction

Education of health professions students is demanding and challenging because of high academic loads and clinical requirements aimed at providing evidence-based knowledge, skills, and competencies to meet patients’ needs and expectations while ensuring high quality of patient care. However, they are frequently faced with text anxiety that negatively affects their cognitive abilities and physical and emotional wellbeing [1]. Test anxiety is a multidimensional construct that consists of affective, physiological, cognitive, and behavioral components. It is a form of anxiety that arises before and during tests and examinations. The students with test anxiety may experience physical symptoms (headache, sweating, rapid heartbeat, etc.), emotional symptoms (stress, fear, disappointment, helplessness, etc.), and cognitive symptoms (negative thinking, difficulty concentrating, etc.) [2,3,4].

Researchers have reported the prevalence of text anxiety among health professions students ranging from 25% to 65% [4,5,6,7]. It is documented that test anxiety is experienced by nearly all students, however, in some students, it can impair their motivation, concentration, learning, and performance in the tests and even can lead to dropping out of school [3,8]. Culler et al. claimed that students with increased test anxiety demonstrated poor study skills and low academic performance [9]. Sansgiry et al. reported that test anxiety was negatively correlated with the ability of students to understand the course material/manage academic load and the ability to prepare for the examinations [10].

A large body of evidence shows that many factors are related to text anxiety among students in different parts of the world. These factors include female gender, low previous grade point average (GPA), maternal educational level, extensive course loads, poor study skills, lack of physical exercise, fear of failing the course, family responsibilities, poor time management skills, lack of social support, psychological distress, and low self-esteem [1,5,11,12,13]. Recently, Gilavand et al. reported severe test anxiety in 9.8%, moderate test anxiety in 32.7%, and mild test anxiety in 37.5% of dental students in Iran [14]. Latas et al. also reported a moderate level of test anxiety among medical students with female students demonstrating significantly higher test anxiety than male students in Belgrade, Serbia [2]. The analysis of data by Aziz and Serafi revealed that more than of half of female medical students had text anxiety in Makkah, Saudi Arabia [15]. Another study in the country showed test anxiety in 65% of medical students with a predilection for female students [5]. There is also evidence of a relationship between test anxiety and under-performance in objective structured clinical examinations among pharmacy students in Saudi Arabia [16]. A recent study observed high test anxiety in more than 50% of medical and dental students in Pakistan [17]. However, the literature is limited regarding the evaluation of test anxiety among medical, dental, and pharmacy students in their undergraduate programs.

An investigation of test anxiety and its related demographic variables (gender, types of students, class year, parental education, and family income) and GPA in health professions students is of particular importance given its high prevalence and adverse effects on academic performance [9,15,17]. The study of the relationship of demographic and academic factors with test anxiety will provide an indication of student groups who are at increased risk of text anxiety. Therefore, the study data can help academia design preventive strategies to minimize test anxiety and improve learning among health professions students and update medical curricula. It was hypothesized that demographic factors and academic performance are significantly related to text anxiety among health profession students. The study aimed to assess test anxiety and its relationship with demographic and academic factors among undergraduate medical, dental, and pharmacy students in Dammam, Saudi Arabia.

## 2. Materials and Methods

The population in this cross-sectional study included undergraduate male and female medical, dental, and pharmacy students from Imam Abdulrahman Bin Faisal University (IAU), Dammam. The study was conducted from September to November 2019. All undergraduate health professions students and those who provided their written consent form were eligible to participate in the study. The students were selected using a convenience sampling technique.

A self-administered questionnaire was used for data collection. Demographic information of study participants was inquired in the first section of the questionnaire. The second section included Test Anxiety Inventory (TAI) developed by Charles D. Spielberger [18]. The TAI consists of 20 statements in which respondents were asked to mention the level of anxiety they experienced before, during, and after taking tests/examinations. Each item of the TAI uses a 4-point Likert scale (1) almost never, (2) sometimes, (3) often, and (4) almost always. The score of TAI ranges from 20 to 80 [18]. The validity and reliability of TAI were confirmed in previous studies [18,19]. In this study, Cronbach’s alpha of TAI was calculated and internal consistency was satisfactory (Cronbach’s α = 0.887). The English version of the questionnaire was used among students because the medium of instruction is English in IAU. The questionnaire was pre-tested among 30 students and their data were not included in the study. The pre-testing was conducted to evaluate readability, comprehension, ease, and time of completion of the survey. No changes were required in the questionnaire after pre-testing.

Ethical approval was obtained from the Deanship of Scientific Research IAU. Administrations in the College of Medicine, College of Dentistry, and College of Pharmacy approved the collection of data in their institutions. After obtaining informed consent, paper-based, self-administered questionnaires were disseminated to 1098 students in their classes before examinations. The participants received information about the study including its purpose and objective, and they were assured of anonymity, privacy, and confidentiality of their responses. Most students returned the completed questionnaires after 15–20 min. Ethical principles of the Declaration of Helsinki were followed during the conduct of the study.

The normality of the TAI score was evaluated, and accordingly non-parametric tests were performed. What demographic factors are related to test anxiety among health profession students? This research question was investigated by evaluating the relationship between test anxiety (TAI score) and demographic factors (gender, class year, parent education, and family monthly income) by using the Mann–Whitney U test and Kruskal–Wallis test. Mann–Whitney U test was performed to compare mean score of TAI in male and female students and students with low GPA versus high GPA. The Kruskal–Wallis test was performed to compare the mean score of TAI in medical, dental, and pharmacy students and in three categories of students based on monthly family income and parental education. The null hypothesis of the study: there is no statistically significant relationship between predictor variables (demographic factors and GPA) and the response variable (text anxiety). Therefore, a multivariate analysis was performed to evaluate the predictors of text anxiety after controlling for other study variables. Data analyses were performed by using Statistical Package for Social Sciences Version 22.0 (IBM SPSS Statistics for Windows, IBM Corp., Armonk, NY, USA). A *p*-value < 0.05 was considered statistically significant.

## 3. Results

Of the total 1098 participants, 878 returned completed questionnaires and the response rate of the study was 79.9%. The study population consisted of 54.9% of females and 45.1% of males with a mean age of 21.45 (SD 1.51). The study included 46.4% of dental, 33.3% of medical, and 20.4% of pharmacy students. The majority of the students belonged to senior classes (74%) and had college/university educated fathers (64.5%) and mothers (56.9%) (Table 1). The mean TAI score of the sample was 43.17 (SD = 10.58).

Analysis of the distribution of students’ responses to each item of the TAI scale showed that 33% of students mentioned that they almost always “wished examinations did not bother so much” and this was the most frequently reported response. This was followed by 15% of students who almost always stated that “thinking about the grade in a course interferes with work on tests”. Whereas only 6.4% of the students mentioned they almost always “seemed to defeat themselves while working on important tests” and 6.8% almost always “felt very jittery when taking an important test.”

The study found statistically significant differences in the mean scores of TAI among dental (44.15 ± 0.48), medical (41.64 ± 1.31), and pharmacy students (43.44 ± 9.29) (*p* = 0.003). The participants with high GPA (mean TAI = 41.91 ± 10.43) demonstrated significantly lower test anxiety than those with low GPA (mean TAI = 44.05 ± 10.67) (*p* = 0.006). In addition, Spearman’s correlation test also found statistically significant negative correlation between test anxiety and GPA (rho = −0.10, *p* = 0.003). No statistically significant relationships between test anxiety and gender, academic year, parent education, and monthly income were found in the study (Table 2).

Further analysis of data showed significantly higher test anxiety in dental students (44.15 ± 0.48) than medical students (41.64 ± 1.31) (*p* = 0.005). Table 3 shows the results of the multivariate analysis. Academic grade in the previous year (GPA) remained a statistically significant factor associated with test anxiety (B = −2.83, *p* = 0.003) after controlling for gender, health professions students, academic year, parental education, and family income.

## 4. Discussion

The present study investigated test anxiety and its relationship with demographic and academic factors among health professions students in Saudi Arabia. The study showed that test anxiety was common (mean TAI score = 43.17 ± 10.58) in our sample of students. This finding is similar to the results of previous studies which reported mean scores of TAI among university students in Canada (43.00 ± 14.47) [1] and Turkey (39.44 ± 11.34) [12]. Likewise, previous studies which employed the TAI scale demonstrated similar results in medical students in Belgrade [2] and dental students in Iran [14]. Conversely, test anxiety evaluated using the Westside Test Anxiety Scale (WTAS) in studies of medical [6,20] and pharmacy [21] students indicated high levels of test anxiety which interfered with their motivation, learning, and academic performance.

In Saudi Arabia, a previous study demonstrated alarming levels of stress (70.90%), anxiety (66.4%), and depression (69.90%) among medical and dental students [22]. Another study also reported high prevalence (47.0%) of depressive symptoms in health professions students in the country [23]. Many reasons have been cited in the literature for increased psychological symptoms among medical students, which include financial problems, parental expectations, excessive workload, language difficulties, lack of time, lack of relaxation, ineffective teaching techniques, and poor testing/examination environments [15,23]. Aziz and Serafi reported test anxiety in 53.04% and psychological distress in 82.50% of medical students in Saudi Arabia and showed a significant positive correlation between test anxiety and psychological distress [15]. Additionally, it was reported that excessive course load (odds ratio (OR) = 6.13), lack of study plan (OR = 2.4), poor social support (OR = 3.6), and psychological distress (OR = 2.68) were significantly associated with test anxiety [6]. High occurrence of psychological disorder and its association with test anxiety including similar contributing factors may explain the reasons for increased test anxiety in the present study.

The literature indicates a statistically significant association between the presence and severity of depressive symptoms and female gender in health professions students in Saudi Arabia [23]. In a previous study, gender was found to be a significant factor associated with test anxiety and female students were 3.25 times more likely than male students to show symptoms of test anxiety [6]. On the contrary, test anxiety did not statistically differ in male and female students in the present study. This is similar to the findings reported by Gilavand et al. [14] in Iran and Latas et al. [2] in Serbia.

The results of the present study demonstrated that students with low GPA demonstrated significantly greater test anxiety than those with high GPA. In addition, it was found that academic grade in previous year was significantly and negatively associated with test anxiety in our sample of students. In Sudan, Sideeg also observed a significant negative correlation (Pearson Correlation Coefficient (r) = −0.476) between test anxiety and academic achievement measured by GPA [24]. Likewise, a significant negative correlation between test anxiety and GPA was reported by Balogun et al. among university students in Nigeria [25]. A study by Culler et al. showed a significant reduction in GPA associated with test anxiety among college students. The authors also reported poor study skills among students with high test anxiety [9]. In contrast, a recent study of medical students by Tsegay showed that high GPA was significantly associated with increased risk of test anxiety in Ethiopia [6]. The students may also have difficulty in time management when dealing with extensive coursework, which is related to test anxiety [5,13]. It is also known that test anxiety can impair concentration and memory and can increase distress, which negatively impacts understanding, learning abilities, and hence academic performance [3].

In the present study, statistically significant differences were observed in test anxiety among health professions students with dental students exhibiting the highest test anxiety, followed by pharmacy and then medical students. The findings of a study in Pakistan showed higher test anxiety in dental than medical students and there were significantly higher odds of high test anxiety in dental compared with medical students (adjusted odds ratio = 1.54, *p* = 0.022) [17]. A study of health professions students by AlFaris et al. from Riyadh, Saudi Arabia showed the highest prevalence of depressive symptoms in dental students (51.6%), followed by medical students (46.2%), and applied medical sciences students (45.7%) [23]. The high prevalence of psychological problems in dental students and their significant correlation with test anxiety may account for increased test anxiety in our sample of dental students [15,23]. In Saudi Arabia, students with higher GPA frequently opt for medical colleges than dental colleges and the existence of an inverse correlation between GPA and test anxiety may explain the phenomenon of greater test anxiety in dental students. In addition, dental programs involve more training of dexterity skills which may increase dental students’ test anxiety.

Students with debilitating test anxiety may be unable to prepare for and perform in their examinations due to lack of awareness of their test anxiety and ineffective coping strategies [26,27]. Reduction in test anxiety can be achieved by employing techniques of stress management, systematic desensitization, cognitive behavior therapy, progressive muscle relaxation, and psychoeducation [27,28,29]. It was reported that 93% of medical students and physicians passed the United States Medical Licensing Examination (USMLE) or a specialty board test after receiving behavioral management of their test anxiety which resulted in failing their examination or tests before treatment. Behavioral management of test anxiety included desensitization, progressive muscle relaxation, self-control triad, behavioral rehearsal, and psychoeducational elements [27]. Previous experimental studies of medical and pharmacy students demonstrated a significant reduction in test anxiety and psychological distress, and improvement in motivation and GPA after psychological intervention, which included psychoeducation, relaxation therapy, and systematic desensitization [28,30].

A recent study of medical and dental students reported a significant relationship between parental education and test anxiety [17]. This is contrary to the findings of the present study where parental education showed no significant influence on text anxiety. In the present study, no statistically significant differences were observed among students with low, middle, and high family incomes. This is in agreement with the results of a previous study where a not significant relationship was found between family income and text anxiety [17].

Generally, junior and senior students in health professions differ in their exposure to different courses (pre-clinical and clinical), teaching and assessment strategies, and patient management [15]. Albeit understandable, junior students demonstrate slightly lower test anxiety than senior students who undergo more rigorous clinical training. In the present study, lower test anxiety was found in junior than senior students; however, the differences were not statistically significant. This finding is consistent with previous studies where preclinical year students showed lower test anxiety than clinical year students with no significant differences [15,17].

To the best of our knowledge, this is the first study that assessed test anxiety and its associated factors among medical, dental, and pharmacy students. The study filled the knowledge gap regarding test anxiety in health profession students, particularly in the Middle Eastern context. Nevertheless, limitations in this study include the use of self-reported data which can compromise the validity of the study. Moreover, data were collected from health professions students from a public university in Dammam, hence care should be exercised when generalizing the study results to students in other institutions in the country. In the future, large multicenter studies should be conducted to investigate test anxiety, a variety of influencing factors such as satisfaction with the faculty, study skills, time management skills, exam methods, exam types, and techniques for its management.

## 5. Conclusions

The study showed that test anxiety was common among undergraduate health professions students. Being a dental student and having a low GPA were significantly related to high test anxiety. Additionally, there was a statistically significant negative correlation between test anxiety and GPA. The investigation of test anxiety in health professions students has important implications in educational settings affecting motivation, psychological health, learning, and academic performance. Early detection and intervention of test anxiety should be performed to improve learning and academic performance among health professions students. Therefore, students, faculty, and counselors should adopt measures to prevent negative consequences of test anxiety on educational experiences, learning processes, and academic performance.

## Figures and Tables

**Table 1 behavsci-12-00098-t001:** Sociodemographic characteristics of medical and dental students (N = 878).

Study Variables	N (%)
Gender	
Male	396 (45.1)
Female	482 (54.9)
Students	
Dental	407 (46.4)
Medical	292 (33.3)
Pharmacy	179 (20.4)
Academic year:	
Junior students (2nd and 3rd years)	228 (26)
Senior students (4th, 5th, and 6th years)	650 (74)
Academic grades in the previous year (N = 683):	
≤4.32 GPA	342 (50.1)
>4.32 GPA	341 (49.9)
Father’s Educational level:	
No education	24 (2.7)
School education	288 (32.8)
College/university education	566 (64.5)
Mother’s Educational level:	
No education	40 (4.6)
School education	338 (38.5)
College/university education	500 (56.9)
Monthly family income (N = 653):	
Low income (less than 10,000 SR)	81 (9.2)
Middle income (10,000–20,000 SR)	257 (29.3)
High income (more than 20,000 SR)	315 (35.9)

**Table 2 behavsci-12-00098-t002:** Bi-variate analysis: Relationship between test anxiety and demographic factors among students.

Study Variables	TAI Score Mean ± SD	*p*-Value
Gender		0.555
Male	42.79 ± 10.17	
Female	43.48 ± 10.91	
Health Professions students		0.003 *
Dental	44.15 ± 0.48	
Medical	41.64 ± 1.31	
Pharmacy	43.44 ± 9.29	
Academic year		0.182
Junior students (2nd and 3rd years)	42.43 ± 10.05	
Senior students (4th, 5th, and 6th years)	43.43 ± 10.76	
Academic grades in the previous year		0.006 *
≤4.32 GPA	44.05 ± 10.67	
>4.32 GPA	41.91 ± 10.43	
Father’s Educational level		0.237
No education	46.21 ± 8.23	
School education	43.17 ± 9.73	
College/university education	43.04 ± 11.07	
Mother’s Educational level		0.646
No education	42.97 ± 9.05	
School education	43.02 ± 9.73	
College/university education	43.29 ±11.25	
Monthly family income		0.101
Low income	43.59 ± 10.75	
Middle income	44.16 ± 9.99	
High income	42.60 ± 11.04	

* Statistically significant.

**Table 3 behavsci-12-00098-t003:** Multivariate analysis: Relationship between test anxiety and demographic factors among students.

Study Variables	TAI Score		*p*-Value
	Unstandardized Coefficients B	Std. Error	
Gender	1.43	0.95	0.134
Health professions students	−0.75	0.53	0.156
Academic year	−0.13	0.99	0.896
Academic grades in the previous year	−2.83	0.94	0.003 *
Father’s Educational level	−0.52	0.49	0.282
Mother’s Educational level	0.52	0.46	0.260
Monthly family income	−0.74	0.46	0.110

* Statistically significant.

## Data Availability

The data can be obtained from the corresponding author on reasonable request.

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
