# Peer review of "Test Anxiety and Related Factors among Health Professions Students: A Saudi Arabian Perspective"

_behavsci, 2022, doi:10.3390/bs12040098_

Round 1

Reviewer 1 Report

Great deal of content in discussion should be moved to conclusion. Conclusion should tell what we should do now that we know this. Conclusion discusses implications for practice.

Author Response

The authors are grateful to the respected reviewer for highly encouraging comments.  Reviewer’s comment addressed and important findings of the study and its implications described in the conclusion. Please see conclusion section of the manuscript.  

Reviewer 2 Report

Thank you for the invitation to review this article.

The study presented by the authors consists of the evaluation of test anxiety and its relationship with demographic factors among health sciences students in Saudi Arabia.

Standardized tests have been used to evaluate anxiety before exams and different variables are assessed, such as the grade they are doing and the grade point average (GPA).

The study provides results that may be of interest to readers of the journal, so I think it can be published. However, I mention a few small details to improve the article:

1) In the abstract I suggest not to put the acronym "GPA" before writing its meaning.

2) The introduction could delve deeper into the anxiety test, developing a greater theoretical framework with previous studies.

3) It would be advisable to write research questions about the study to favor the narration of the results.

Author Response

The study provides results that may be of interest to readers of the journal, so I think it can be published. However, I mention a few small details to improve the article:

1) In the abstract I suggest not to put the acronym "GPA" before writing its meaning.

Authors’ response to reviewer’s comment. Reviewer’s comment addressed. Please see abstract section of the manuscript.

2) The introduction could delve deeper into the anxiety test, developing a greater theoretical framework with previous studies.

Authors’ response to reviewer’s comment. Detailed review of previous similar studies on test anxiety was presented in the introduction section.

3) It would be advisable to write research questions about the study to favor the narration of the results.

Authors’ response to reviewer’s comment. Reviewer’s comment addressed. Please see last paragraph of material and methods section of the manuscript.

Reviewer 3 Report

The article deals with an important topic, which is test anxiety in students of various medical specialties. The authors also analyze the influence of demographic factors on the level of the studied variable. The text is well written, the analyses are correctly conducted, and the conclusions are properly derived.
I have a minor comment regarding the discussion conclusion regarding the motivation for the analyses."

Because test anxiety is known to affect academic performance, therefore, the relationship of parental income with test anxiety was investigated in the present study”.

This conclusion does not seem legitimate and needs improvement.

Author Response

Authors are extremely grateful to respected reviewer for his/her satisfaction with the quality of our paper. Reviewer’s comment addressed. Please see discussion section of the manuscript.

Reviewer 4 Report

The paper  is clear and easy to read. The main question addressed by the research is interesting.

The subject contributed by the authors in comparison to other publications and other published materials  evaluate test anxiety and its relationship with demographic factors among undergraduate medical, dental, and pharmacy students in Dammam, Saudi Arabia. 
There were statistically significant differences in the mean scores of TAI (Test Anxiety Inventory) among dental , medical and pharmacy  students. 

The conclusions are consistent with the evidence and arguments presented. They address the main question posed. This study showed that students with low GPA (Academic grade in previous year) demonstrated high test anxiety. Therefore, the students with high test anxiety should be the target of preventive strategies and early detection and intervention of test anxiety should be performed to improve learning and academic performance among health professions students. 

Areas of the strength of this paper:
The authors evaluate test anxiety and its relationship with demographic factors among undergraduate medical, dental, and pharmacy students in Dammam, Saudi Arabia.

Areas of  weakness of this paper:
In addition to the fact that the subject is not original, the authors should indicate how to proceed, once the results have been obtained, although this would surely be part of another future investigation.

Author Response

Authors are extremely grateful to respected reviewer for his/her satisfaction with the quality of our paper. The weaknesses of our manuscript and future investigation were discussed in the discussion section of the manuscript. Please see last paragraph of discussion section of the manuscript.